# The effects of high-frequency repetitive transcranial magnetic stimulation on negative symptoms in schizophrenia patients: A systemic review and meta-analysis

Boxing Wang[1�], Xinyue Zhu[2�], Ling Chen[3], Shuyun Liu[4]*, Chengshi Wang[5]*

**1** West China Medical school, Sichuan University, Chengdu, China, **2** Department of Endocrinology and Metabolism, Center for Diabetes and Metabolism Research, West China Hospital, Sichuan University, Chengdu, China, **3** Institute of Taoism and Religious Culture, Sichuan University, Chengdu, China, **4** Department of General Surgery and NHC Key Laboratory of Transplant Engineering and Immunology, Frontiers Science Center for Disease-related Molecular Network, West China Hospital of Sichuan University, Chengdu, China, **5** Department of Endocrinology and Metabolism, Laboratory of Diabetes and Metabolism Research, West China Hospital, Sichuan University, Chengdu, China

☻ These authors contributed equally to this work.
* wangchengshi@wchscu.cn (CW); shuyunkunming@163.com (SL)

## Abstract

### Objective

The research on high-frequency repetitive transcranial magnetic stimulation (rTMS) is very ambiguous, particularly the variability in treatment parameters and the need for standardized protocols, and its effectiveness have hardly been provided conclusive evidence. This systematic review and meta-analysis aimed to determine the effects of high-frequency rTMS on negative symptoms in schizophrenia patients.

### Methods

Six databases (PubMed, Embase, Web of Science, SinoMed, PsycINFO and MED-LINE) were searched from inception to March 2024. Relevant data were extracted and analysed by two independent investigators using Cochrane RevMan software (version 5.3) and R software (version 4.3.1).

### Results

A total of 17 Randomized Controlled Trials (RCTs) were included in the current meta-analysis. Compared with control treatment, active rTMS showed advantage in treating negative symptoms of schizophrenia [standardized mean difference (SMD): −0.22, 95% confidence interval (CI): −0.38, −0.05, $P=0.009$; $I^2=9\%$]. Sub-group analysis revealed that rTMS with a total treatment course of more than 15 sessions [SMD: −0.31 (95% CI: −0.49, −0.13), $P=0.0007$; $I^2=0\%$], active rTMS with

**Data availability statement:** All relevant data are within the manuscript and its Supporting Information files.

**Funding:** The author(s) received no specific funding for this work.

**Competing interests:** The authors have declared that no competing interests exist.

**Abbreviations:** rTMS, repetitive Transcranial Magnetic Stimulation; RCTs, Randomized Clinical Trials; PANSS, Positive and Negative Syndrome Scale; SMD, Standardized mean Difference; CI, Confidence Interval; DLPFC, Dorsolateral Prefrontal Cortex; DSM, Diagnostic and Statistical Manual of Mental Disorders; ICD, International Statistical Classification of Diseases and Related Health Problems; CDSS, Calgary Depression Scale for Schizophrenia; SANS, Scale for the Assessment of Negative Symptoms.

a frequency of 20 Hz [SMD: −0.40 (95% CI: −0.67, −0.13), $P = 0.004$; $I^2 = 0\%$], and active rTMS targeting the dorsolateral prefrontal cortex (DLPFC) [SMD: −0.25 (95% CI: −0.40, −0.09), $P = 0.003$; $I^2 = 0\%$] were significantly effective in terms of reducing negative symptoms.

## Conclusions

The effect of rTMS on negative symptoms is influenced by rTMS parameters. We suggest that researchers should focus on changing the frequency, location and length of treatment to improve negative symptoms of schizophrenia.

## Introduction

Schizophrenia is a serious mental disorder that affects approximately 1% of the global population [1]. The diagnosis of schizophrenia is typically based on clinical evaluations. The disorder is characterized by several positive symptoms, such as hallucinations and delusions, as well as negative symptoms, such as a lack of emotion, decreased joy or motivation, delayed speech, social withdrawal, and difficulty initiating and sustaining activities [2]. Furthermore, individuals with schizophrenia also suffer from other cognitive deficits, including impairments in working memory, executive function, and processing speed [3,4].

Negative symptoms are one of the core features of schizophrenia and are associated with long-term morbidity and poor functional outcomes. Recent clinical studies have suggested that up to 60% of patients may exhibit prominent negative symptoms that necessitate treatment [5–7]. However, antipsychotic treatments, such as antipsychotic drugs with dopamine D2 antagonists or partial D2 agonists, are not optimal methods [8–10]. While these drugs may be effective for treating some positive symptoms, they often fail to address negative symptoms and can lead to side effects such as weight gain and metabolic syndrome [11,12]. Furthermore, the clinical diagnosis of negative symptoms is challenging because schizophrenia patients may not recognize the impact of negative symptoms and are unlikely to report them [13].

In recent years, several studies have demonstrated that high-frequency rTMS have effect on reducing negative symptoms in schizophrenia patients [14–16]. Furthermore, several meta-analyses and reviews have also provided evidence supporting the use of high-frequency rTMS for treating these symptoms [17–20]. However, the frequency, location and sessions of high-frequency rTMS remain controversial [18,19,21–22]. Recently, high-frequency rTMS also has therapeutic implications for depression [23,24], but studies have suggested that rTMS may not have a significant benefit on positive symptoms of schizophrenia [25,26]. Therefore, we conducted this systematic review and meta-analysis to assess the effectiveness of high-frequency rTMS for reducing negative symptoms among schizophrenia patients, and to assess the effectiveness for depression and positive symptoms of schizophrenia.

## Materials and methods

### Registration and protocol

This systematic review and meta-analysis were registered in the International Prospective Register of Systematic Review (PROSPERO) trial registry (CRD42023450243) [27]. Ethical approval and patient consent were not required as all the analyses were based on previously published studies.

### Search strategy and selection criteria

The systematic search was conducted on the electronic database including PubMed, EMBASE, Web of Science, SinoMed, PsycINFO and MEDLINE database for articles published from inception to July, 2023 with the following key-words: "rTMS", "repetitive transcranial magnetic stimulation", "Schizophrenia", "high-frequency" and "Negative Symptoms". The reference lists of retrieved studies and relevant reviews were hand-searched, and the process mentioned above was repeatedly performed for ensuring that all eligible studies were included.

Inclusion criteria are presented as follows: (1) study design is randomized controlled trial (RCT); (2) the frequency was > 1 Hz rTMS; (3) patients diagnosed with schizophrenia or schizoaffective disorder diagnosed according to standardized criteria such as Diagnostic and Statistical Manual of Mental Disorders (DSM), International Statistical Classification of Diseases and Related Health Problems (ICD) or MiniInternational Neuropsychiatric Interview. The severity of negative symptoms with schizophrenia were assessed with PANSS [28,29] and the SANS [30]. The severity of positive symptoms with schizophrenia PANSS, and depression with Schizophrenia were assessed with the Calgary Depression Scale (CDSS) [31]; (4) without comminating other psychiatric diseases.

### Data extraction

The data were extracted by two individuals independently: (1) author, title, journal, publication year and study design; (2) the PANSS, the SANS and the CDSS rating scale values; (3) demographic and clinical characteristics were also extracted including (age, age of onset, duration of illness, gender distribution, antipsychotic medication, rTMS condition, frequency of treatment, follow up time and outcome measurements). In case of missing data, we tried to contact the author to obtain the data. Alternatively, we performed subgroup analyses based on different data to ensure stability of the results.

### Outcome measures

The primary outcome was the efficacy of high-frequency rTMS group versus control group on negative symptoms in patients diagnosed with schizophrenia. PANSS or SANS were consistently used as outcomes measurement. Severity of negative symptoms is positively correlated with level of score. The secondary outcomes were narrative analysis of the results from articles not included in the primary analysis, PANSS positive, and CDSS were used to evaluate the synergistic effect of high-frequency rTMS.

### Quality assessment and risk of bias

The Cochrane risk of bias was used by two investigators independently and with the differences resolved by discussion with all authors. The domain-based evaluation recommended by the Cochrane Handbook for Systematic Reviews of Intervention were used to address the following domains: bias arising from the randomisation process, bias due to deviations from intended interventions, bias due to missing outcome data, bias in measurement of the outcome and bias in selection of the reported result [32]. Fig 1b and 1c were made by using Review Manager Software Version 5.3 (The Cochrane Collaboration, Software Update, Oxford, UK).

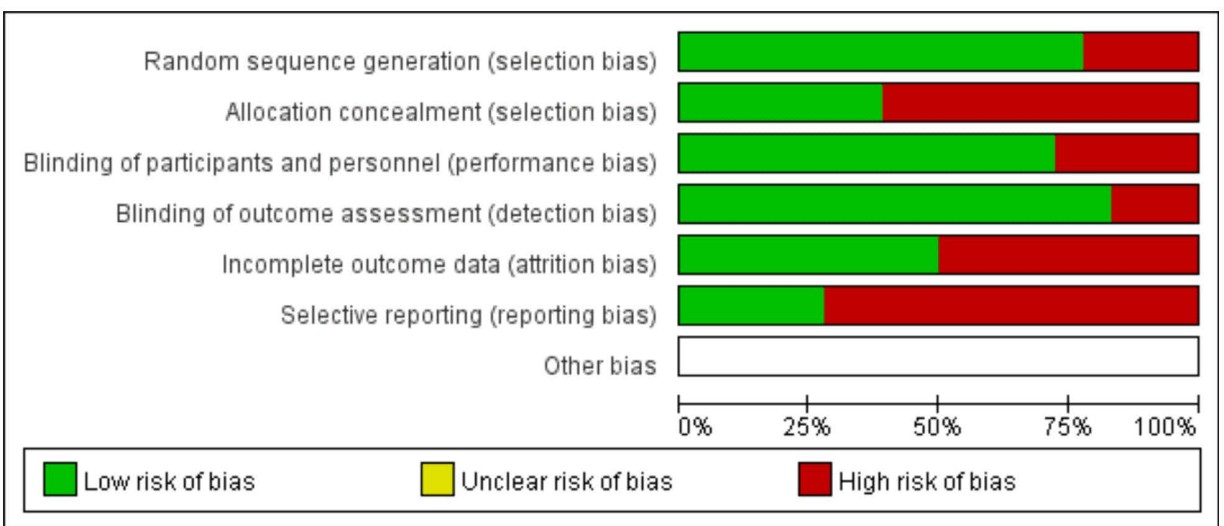

**Fig 1.  a. Flow diagram of the study search and selection process, b. Risk of bias summary, c. Risk of bias graph.**

## Statistical analysis

Cochrane RevMan software (version 5.3) and R software (version 4.3.1) were used to conduct the statistical analysis. Given the anticipated heterogeneity in global data, a random-effects model was employed to analyse. Cohen's d standardized mean differences (SMD) were computed to calculate the random-effect meta-analysis with inverse variance weighting. In this meta-analysis, 95% confidence intervals (CIs) were calculated for categorical and continuous outcomes, significance level was set as $P<0.05$. Heterogeneity among studies was determined using the $I^2$ test, with a $P$-value less than 0.1 and an $I^2$ – value over 50%, suggesting the presence of significant study heterogeneity. Sensitivity analysis was performed for evaluating the influence of a single study on the overall estimate by omitting one study. Publication bias of the primary outcome was evaluated using Funnel plots and Egger's test. Subgroup analyses were performed to assess potential confounding effects of heterogeneity. Four subgroup analyses were also conducted for further investigation: (1) applied 15 continues stimulation sessions or more 15 stimulation sessions; (2) the frequency of rTMS was 10 Hz; (3) the frequency of rTMS was 20 Hz; (4) the location of stimulate was DLPFC. The difference between groups was assessed using a $P$-value, with a threshold of $P<0.05$ indicating a statistical significance.

## Results

### Literature search

A systematic search was performed according to the PRISMA flow diagram. A total of 59 articles were identified after reviewing the literature, and 17 studies were ultimately included. The reasons for exclusion were as follows: two studies focused on the efficacy of high-frequency rTMS for preventing cognition, two studies lacked endpoint results, four studies used drug combinations, two studies investigated the relationship between smoking and high-frequency rTMS, and several other studies focused on different topics (auditory verbal hallucinations, low-frequency rTMS, and mechanical repetitive transcranial magnetic stimulation). Fig 1a shows the PRISMA flow chart for the study selection process. The characteristics of the included RCTs are summarized in Table 1. Seventeen double-blind RCTs (n=911) were included. Xiu et al 2020 [33] contain 10 Hz and 20 Hz as treatment. Thirteen RCTs involved stimulation in the DLPFC, and the frequency of treatment was five sessions a week. Twelve RCTs had more than 15 sessions in total. When the frequency of rTMS ranged from 5 Hz to 20 Hz, eight and six studies had frequencies of 10 Hz and 20 Hz, respectively. Other details are presented in Table 2.

### Quality assessment

Fourteen RCTs (14/18, 77.78%) reported an adequate method of random sequence generation. Other risk of bias domains was rated as "unclear" in all RCTs (18/18, 100%). Based on the Cochrane risk of bias tool, the overall quality of the outcomes ranged from "low risk" (63/108, 58.33%) to "high risk" (45/108, 41.67%). The risk of bias graph and risk of bias summary are shown in Fig 1.

### Negative symptoms

The primary outcome of this study was the effect of high-frequency rTMS versus control-assisted rTMS on negative symptoms in patients with schizophrenia. The 15 RCTs in our meta-analysis that included PANSS-negative scores suggested that high-frequency rTMS is beneficial for treating negative symptoms in schizophrenia patients. Additionally, 10 studies including SANS scores found a stronger effect than the studies using PANSS scores. The PANSS was used to assess negative symptoms, and two frequencies (10 Hz and 20 Hz) were adopted by Xiu et al. PANSS-negative and

**Table 1. Characteristics of included studies.**

| No. | Author | Year | rTMS Group | | | Control Group | | |
|---|---|---|---|---|---|---|---|---|
| | | | Number | Age | AP [c] dosage, chlorpromazine equivalents (mg) | Number | Age | AP † dosage, chlorpromazine equivalents (mg) |
| 1 | Barr et al. | 2012 | 13 | 40.46 (12.21) | 388.89(328.86) | 12 | 47.92(12.78) | 813.49(560.15) |
| 2 | Du et al. | 2022 | 20 | 45.9(10.0) | 323.5(193.1) | 18 | 45.1(10.4) | 341.7(168.7) |
| 3 | Fitzgerald et al. | 2008 | 10 | 37.2 (10.4) | not reported | 10 | 33.2 (9.8) | not reported |
| 4 | Garg et al. | 2016 | 20 | 32.40 (8.44) | 388(110) | 20 | 30.75 (7.90) | 363(129) |
| 5 | Holi et al. | 2004 | 11 | 38.5(10.2) | 1168 | 11 | 34.8(9.8) | 1309 |
| 6 | Kumar et al. | 2020 | 50 | 32.4 (9.20) | not reported | 50 | 30.8(9.34) | not reported |
| 7 | Liu et al. | 2017 | 20 | 30.75(3.9) | not reported | 20 | 31.60(3.7) | not reported |
| 8 | Li et al. | 2017 | 25 | 50. 32 (7. 88) | 378. 64 (194. 10) | 25 | 50. 24 (9. 58) | 385.56 (169.74) |
| 9 | Li et al. | 2018 | 20 | 41.58(6.34) | not reported | 21 | 40.63(6.2) | not reported |
| 10 | Quan et al. | 2015 | 78 | 46.87(7.87) | 411.78(194.19) | 39 | 46.87(9.07) | 438.46(189.58) |
| 11 | Singh et al. | 2020 | 15 | 33.3 (9.8) | not reported | 15 | 29.8 (5.7) | not reported |
| 12 | Wen et al. | 2021 | 26 | 41.4(7.5) | 435.8(302.6) | 26 | 38.8(9.1) | 467.1(267.6) |
| 13 | Wobrock et al. | 2015 | 62 | 36.2 (10.5) | 572 (435) | 64 | 34.9(9.1) | 597 (486) |
| 14 | Xiu et al. [a] | 2020 | 40 | 50.7(9.0) | 416.9(257.6) | 40 | 54.7 (6.4) | 416.9(257.6) |
| 15 | Xiu et al. [b] | 2020 | 40 | 52.0(10.1) | 422.3(231.7) | 40 | 54.7 (6.4) | 416.9(257.6) |
| 16 | Prikryl et al. | 2013 | 23 | 31.60(8.04) | 282.96(231.38) | 17 | 33.94(9.98) | 387.19(272.51) |
| 17 | Huang et al. | 2018 | 30 | 37.43(10.96) | not reported | 30 | 38.00(8.56) | not reported |
| 18 | Lin et al. | 2018 | 30 | 38.9(3.6) | not reported | 30 | 39.2(3.3) | not reported |

AP: Antipsychotic; [a]10 Hz as treatment; [b]20Hz as treatment; [c] Some studies did not calculate the chlorpromazine equivalents which is marked with a hyphen in the table;

**Table 2. Stimulation parameters.**

| Author | Year | Target | Stimulation (Hz) | Density | Sessions | Frequency(day/week) |
|---|---|---|---|---|---|---|
| Barr et al. | 2012 | DLPFC | 20 | 90% | 20 | 5 |
| Du et al. | 2022 | DLPFC | 10 | 110% | 20 | 5 |
| Fitzgerald et al. | 2008 | PFC | 10 | 110% | 15 | 5 |
| Garg et al. | 2016 | Cerebellum | 5/6/7 Hz | 100% | 10 | 5 |
| Holi et al. | 2004 | DLPFC | 10 | 100% | 20 | 5 |
| Kumar et al. | 2020 | DLPFC | 20 | 100% | 20 | 5 |
| Liu et al. | 2017 | DLPFC | 20 | not reported | 24 | 5 |
| Li et al. | 2017 | DLPFC | 10 | 100% | 20 | 5 |
| Li et al. | 2018 | DLPFC | 20 | 100% | 40 | 5 |
| Quan et al. | 2015 | DLPFC | 10 | 80% | 20 | 5 |
| Singh et al. | 2020 | DLPFC | 20 | 100% | 20 | 5 |
| Wen et al. | 2021 | DLPFC | 10 | 110% | 20 | 5 |
| Wobrock et al. | 2015 | DLPFC | 10 | 110% | 15 | 5 |
| Xiu et al. [a] | 2020 | DLPFC | 10 | 110% | 40 | 5 |
| Xiu et al. [b] | 2020 | DLPFC | 20 | 110% | 40 | 5 |
| Prikryl et al. | 2013 | DLPFC | 10 | 110% | 15 | 5 |
| Huang et al. | 2018 | DLPFC | 15 | 100% | 40 | 5 |
| Lin et al. | 2018 | DLPFC | 15 | 100% | 40 | 5 |

PFC: bilateral prefrontal cortex; DLPFC: dorsolateral prefrontal cortex; [a]10 Hz as treatment; [b]20Hz as treatment;

SANS scores were used as endpoints after the end of treatment. Standardized mean differences were calculated, and the random effects model was used to analyse the primary outcomes.

Significant differences were found (SMD = −0.31; 95% CI = −0.49 – −0.14; $P < 0.05$) (Fig 2) between the high-frequency rTMS group and the control group. A total of 791 participants were included in the analysis, and the endpoint was PANSS-negative scores. There was significant heterogeneity among the studies ($I^2 = 32\%$, $P = 0.11$). We analysed SANS as an endpoint; the results showed significant differences (SMD = −0.99; 95% CI = −1.64 to −0.33; $P < 0.05$) (Fig 3), with significant heterogeneity among the studies ($I^2 = 91\%$, $P < 0.05$).

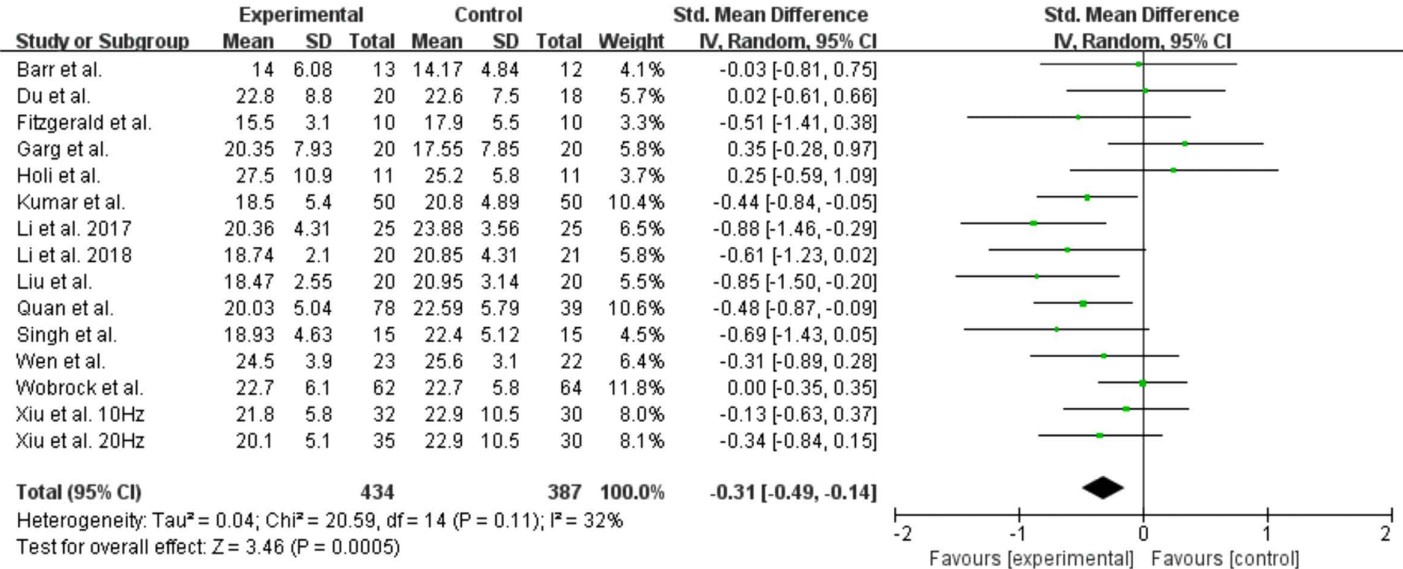

**Fig 2. Forest plot for the meta-analysis of Positive and Negative Syndrome Scale (PANSS) negative symptom scores.**

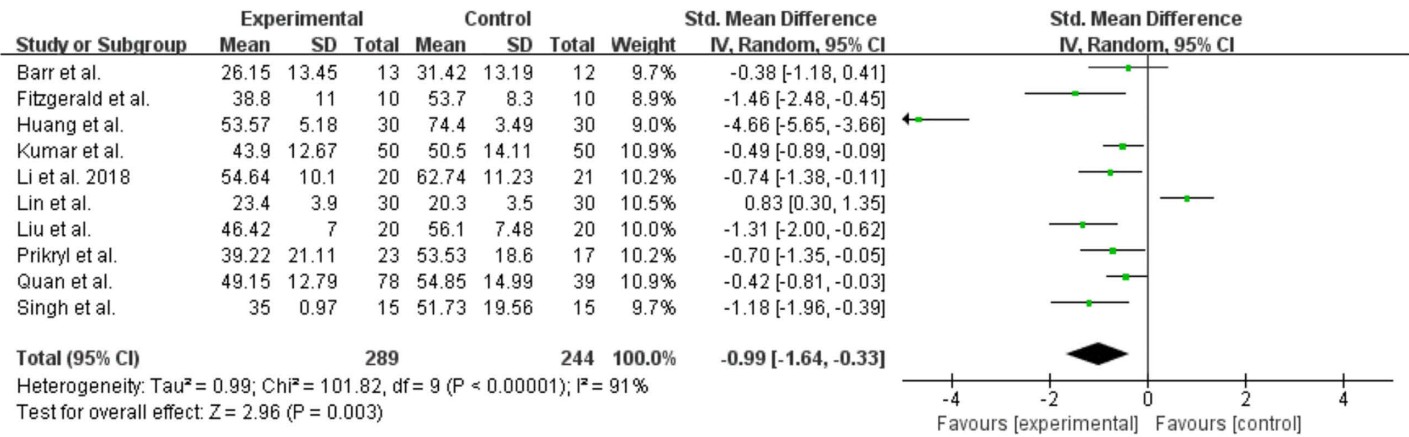

**Fig 3. Forest plot for the meta-analysis of Scale for Assessment of Negative Symptoms (SANS).**

## Subgroup analysis

Standardized mean differences were also calculated, and the random effects model was used to analyse the primary outcomes in Table 3.

We performed subgroup analysis to determine whether 15 sessions or more led to significant differences. The twelve included studies were carefully read. PANSS-negative scores were used to compare the efficacy of active versus control rTMS. In total, 605 participants were included in this subgroup analysis. The significant effects were reported (SMD = −0.40; 95% CI = −0.57 – −0.24; $P < 0.05$), and there was no significant heterogeneity among the studies ($I^2 = 7\%$, $P = 0.38$) (Fig 4a).

Another subgroup analysis was conducted to investigate the effect of the frequency of rTMS. We found that 20 Hz (Fig 4c) (SMD = −0.48; 95% CI = −0.71 – −0.25; $P < 0.05$) had a stronger effect than 10 Hz (Fig 4b) (SMD = −0.26; 95% CI = −0.49 – −0.02; $P < 0.05$). We found significant heterogeneity among the studies using 10 Hz ($I^2 = 33\%$, $P = 0.17$) and no significant heterogeneity among the studies using 20 Hz ($I^2 = 0\%$, $P = 0.65$).

An additional subgroup analysis was performed to determine the effect of the location of the stimulation which is the most place that DLPFC. Thirteen studies were included in this subgroup analysis, and significant differences were observed (Fig 4d) (SMD = −0.34; 95% CI = −0.52 – −0.17; $P < 0.05$). There was low heterogeneity among the studies ($I^2 = 25\%$, $P = 0.19$).

## Secondary outcomes

**Positive symptoms.** The PANSS is widely used as a clinical assessment tool for evaluating the severity of symptoms in individuals with schizophrenia. Fourteen studies reported this secondary outcome; no significant effects (Fig 4e) (SMD = −0.02; 95% CI = −0.16–0.13; $P = 0.81$) were found, and there was no significant heterogeneity among the studies ($I^2 = 2\%$, $P = 0.43$).

**Calgary Depression Scale for Schizophrenia.** The CDSS is a rating scale used to assess depression in individuals with schizophrenia. It was developed specifically to measure depressive symptoms that may be present alongside the core symptoms of schizophrenia. The results of meta-analysis did not reveal any statistically significant differences (Fig 4f) (SMD = −0.09; 95% CI = −0.30–0.13; $P = 0.43$).

## Sensitivity analysis

Sensitivity analysis is one of the most common statistical methods and is used mainly to evaluate the robustness and reliability of combined meta-analysis results. No statistically significant differences were found (Fig 5) (SMD = −0.31; 95% CI = −0.49–0.14; $P = 0.11$).

**Table 3. Subgroup and secondary outcomes.**

| Variables | Studies (subjects) | SMDs (95%CI) | $I^2$ (%) | *P*-value |
|---|---|---|---|---|
| **Subgroup** | | | | |
| More than 15 sessions | 12 (635) | -0.40 [-0.57, -0.24] | NA | <0.05 |
| 10 Hz group | 8 (480) | -0.26 [-0.49, -0.02] | 33 | <0.05 |
| 20 Hz group | 6 (301) | -0.48 [-0.71, -0.25] | NA | <0.05 |
| DLPFC group | 13 (761) | -0.34 [-0.52, -0.17] | 25 | <0.05 |
| **Secondary outcomes** | | | | |
| Positive symptom | 14 (801) | -0.02 [-0.16, 0.13] | NA | 0.81 |
| CDSS | 6 (361) | -0.09 [-0.30, 0.13] | NA | 0.43 |

Abbreviations: CI: confidence interval; SMDs: Standard mean differences; DLPFC: dorsolateral prefrontal cortex; CDSS: Calgary Depression Scale for Schizophrenia; Table: Lists key information about each study, including: authors of the study, number of participants in each study, effect size (SMD): effect size and its 95% CI for each study, weight (%): weight of each study in the meta-analysis.

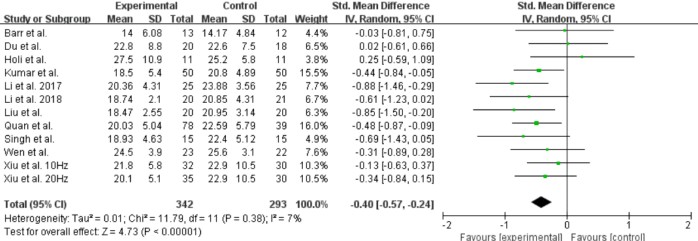

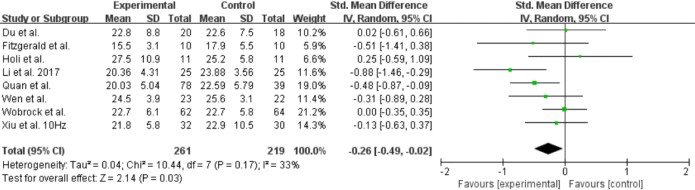

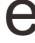

**Fig 4. Forest plots showing primary outcomes in high-frequency repetitive transcranial magnetic stimulation (rTMS) vs control a.** More than 15 sessions, b. 10-Hz group, c. 20-Hz group, d. dorsolateral prefrontal cortex (DLPFC) group. Forest plot for Secondary outcomes **e.** PANSS positive symptom scores and **f.** Calgary Depression Scale for Schizophrenia (CDSS).

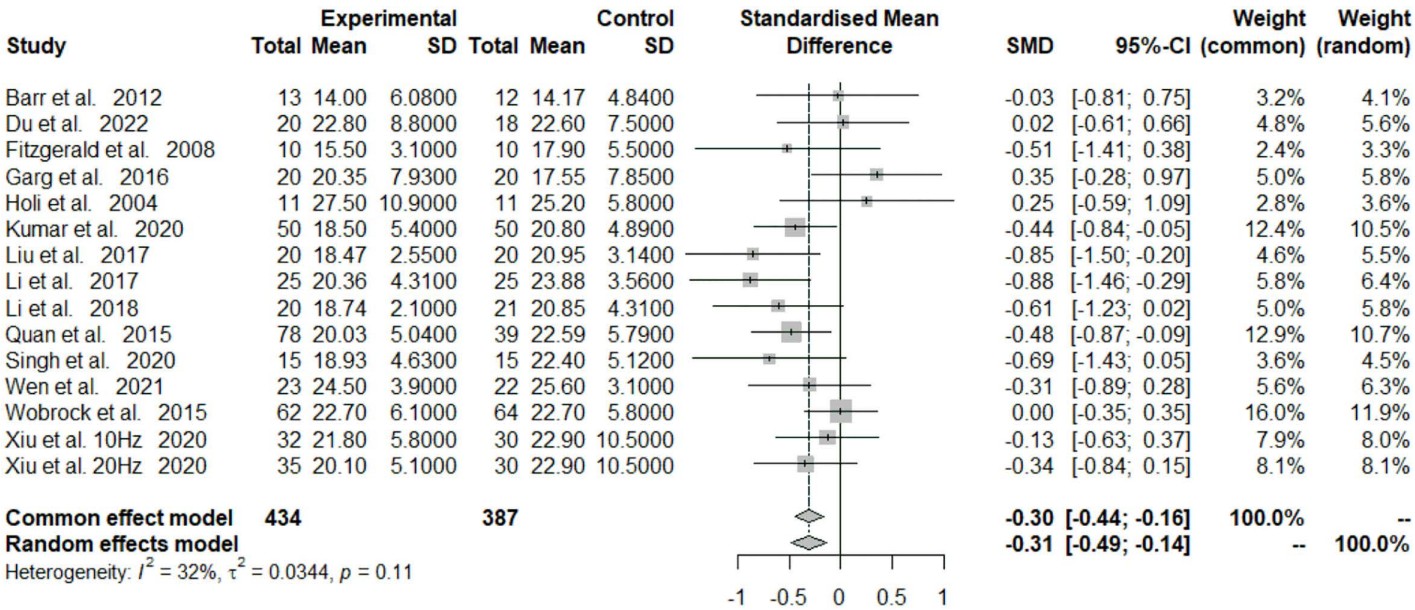

**Fig 5. Forest plot for the sensitivity analysis.**

## Publication bias

Initially, the funnel plot showed no asymmetry (Fig 6). Egger's regression test was performed to quantify the possible amount of bias, and the result was nonsignificant (*P*=0.9605). Taken together, these complementary analyses indicate the absence of publication bias in this meta-analysis.

## Discussion

We conducted a meta-analysis of 17 randomized controlled trials that examined the use of high-frequency rTMS to treat negative symptoms in patients with schizophrenia. Among these studies, 15 included PANSS-negative scores, and 1 included only SANS scores. The SANS is a reliable scale for assessing negative symptoms in patients with schizophrenia [34]. The PANSS-negative scores suggested that high-frequency rTMS is beneficial for treating negative symptoms in schizophrenia patients. Additionally, 10 studies including SANS scores found a stronger effect than the studies using PANSS scores. Although the PANSS and SANS are widely used, their differences may result in variations in baseline data, different frequencies used in the experiments, and different stimulation locations. The PANSS is a more comprehensive assessment tool that covers both positive and negative symptoms as well as general psychopathology, while the SANS focuses exclusively on negative symptoms. Both scales can be used depending on the needs of research or clinical practice, and they are sometimes used in conjunction to obtain a more complete symptom assessment [35]. In the articles we included, the procedures were consistent with the experimental group except for the orientation of the coils or the dummy coils. To further explore the effect of stimulation frequency on negative symptoms in schizophrenia patients, we conducted two subgroup analyses: one with a 10-Hz stimulation frequency (eight studies) and another with a 20-Hz

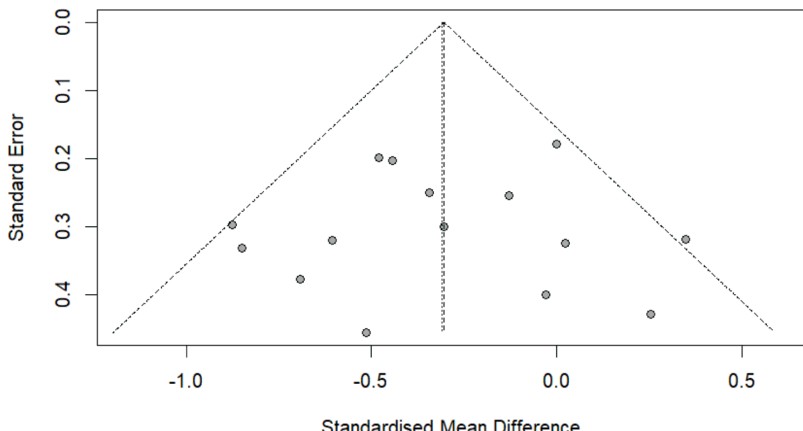

**Fig 6. Funnel plot of high-frequency rTMS effect on negative symptom in schizophrenia.** Each square represents the effect size (e.g., standardized mean difference, SMD) for a single study, and the size of the square is proportional to the weight of the study. The horizontal line represents the 95% confidence interval (CI). The diamonds represent the combined effect sizes of all studies and their 95% CIs.

stimulation frequency (six studies) [19]. Surprisingly, the studies using a 10-Hz frequency showed significant effects ($P<0.05$), but the studies using a 20-Hz frequency showed stronger effects ($P<0.05$). We found that although the effect size of the 20 Hz treatment (−0.40 [−0.67, −0.13]) was greater than that of the treatment of 10 Hz (−0.18 [−0.37, 0.01]), there was an overlap of the two confidence intervals, suggesting that the difference between the two groups may not be statistically significant. Numerous studies have recommended that more than three weeks of treatment may be necessary to achieve more significant effects, which means that 15 sessions or more could yield effective outcomes [36]. Most studies recommend a treatment frequency of five times per week, and the duration of treatment increases with the number of treatments. In the 12 studies with treatment durations longer than three weeks, we observed only slight improvements, which demonstrated that an increase in duration did not result in stronger effects. The safety and tolerability of rTMS at the time of treatment have also been validated in previous studies. [12]

The DLPFC is considered an effective stimulation location [37] and a common target for noninvasive brain stimulation in the treatment of schizophrenia [38,39]. It is closely connected to schizophrenia-spectrum disorders [40,41], negative symptoms and impairments in higher cognitive functions are associated with dysfunction in the DLPFC [42]. We found that 13 studies stimulating DLPFC targeting had significant results. After excluding the study that targeted the cerebellum, the effect size significantly increased, further confirming that the DLPFC is a relevant stimulation target for schizophrenia. While exploring factors associated with negative symptoms in patients with schizophrenia, we found that most studies also reported PANSS-positive and CDSS scores. We found that high-frequency rTMS did not significantly affect these two outcomes' measures ($P=0.81$, $P=0.43$).

The 17 studies included in this analysis had mild heterogeneity, which can be attributed to several factors. First, there was a significant difference in the survival time between baseline and the endpoint, ranging from 14 to 56 days. Second, patients had different baseline characteristics at enrolment. Finally, there were significant differences in the stimulation targets, stimulation parameters (such as intensity, pulse number, and control procedures), and evaluation principles used in different studies, thus highlighting the need to standardize rTMS protocols in the treatment of negative symptoms among patients with schizophrenia.

Admittedly, our meta-analysis has several limitations. First, the heterogeneity of the baseline data, including age, sex, methods, specifications of the equipment, stimulation threshold and staging of illness, may have introduced bias in our results. Garg et al. found a more significant improvement in the control intervention group by targeting the

cerebellum, suggesting that the DLPFC may be an important target for schizophrenia-spectrum disorders [43,44]. Second, the scoring system used herein is more subjective than the other scoring systems, which could introduce bias into the results. Finally, regarding schizophrenia patients with negative symptoms, among the results, we did not find that patients with positive symptoms and depression could benefit from the treatment course of rTMS, which may be further emphasized by the specific efficacy of rTMS on negative symptoms, however, our treatment group was not very abundant, so such results need to be proven by further studies. Therefore, further standardization and uniformity of the procedures must be established between each study. Finally, the missing negative and unpublished data in the original studies may have led to publication bias and skewed our conclusions. Thus, we suggest that robust RCTs with large sample sizes and a standard protocol should be performed in the future to obtain more accurate data and verify our results.

## Conclusions

In conclusion, our systematic meta-analysis provides evidence that the efficacy of high-frequency rTMS on negative symptoms are associate with rTMS parameters. We recommend a frequency of 20 Hz, more than 15 treatment sessions may benefit in the DLPFC. However, further large-scale randomized controlled trials, specific measurement scales for assessing negative symptoms in schizophrenia, standardized rTMS treatment protocols, and consistent outcome measures are needed to confirm the efficacy of rTMS in treating negative symptoms in schizophrenia patients.

Practitioner Points

- Schizophrenia with negative symptoms need to benefit from non-pharmacological treatment modalities.

- The location and frequency of high-frequency repetitive transcranial magnetic stimulation, as well as the scale scored, may affect the treatment effect.

- High-frequency repetitive transcranial magnetic stimulation may benefit patients with schizophrenia who have significant negative symptoms as well as positive and depressive symptoms.

## Supporting information

**S1 Checklist. PRISMA 2020 Checklist.**
(DOCX)

**S1 Table.**
(DOCX)

**S2 Table.**
(DOCX)

## Author contributions

**Conceptualization:** Boxing Wang, Ling Chen.

**Formal analysis:** Boxing Wang, Xinyue Zhu.

**Methodology:** Shuyun Liu.

**Writing – original draft:** Boxing Wang, Xinyue Zhu, Ling Chen.

**Writing – review & editing:** Boxing Wang, Xinyue Zhu, Ling Chen, Shuyun Liu, Chengshi Wang.

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
