## [Decision Letter · Decision Letter 0]

5 Nov 2024

Dear Dr. wang,

Thank you for submitting your manuscript to PLOS ONE. After careful consideration, we feel that it has merit but does not fully meet PLOS ONE’s publication criteria as it currently stands. Therefore, we invite you to submit a revised version of the manuscript that addresses the points raised during the review process.

We look forward to receiving your revised manuscript.

Kind regards,

Giuseppe Lanza, M.D., Ph.D.

Academic Editor

PLOS ONE

https://www.sciencedirect.com/science/article/pii/S0022395621005409?via%3Dihub

https://doi.org/10.3389/fendo.2022.1098325

In your revision ensure you cite all your sources (including your own works), and quote or rephrase any duplicated text outside the methods section. Further consideration is dependent on these concerns being addressed.

3. As required by our policy on Data Availability, please ensure your manuscript or supplementary information includes the following:

Additional Editor Comments (if provided):

Reviewers' comments:

Reviewer's Responses to Questions

**Comments to the Author**

1. Is the manuscript technically sound, and do the data support the conclusions?

Reviewer #1: Yes

Reviewer #2: Partly

2. Has the statistical analysis been performed appropriately and rigorously?

Reviewer #1: Yes

Reviewer #2: No

3. Have the authors made all data underlying the findings in their manuscript fully available?

Reviewer #1: Yes

Reviewer #2: No

4. Is the manuscript presented in an intelligible fashion and written in standard English?

Reviewer #1: Yes

Reviewer #2: No

Reviewer #1: The manuscript investigated the effects of high-frequency repetitive transcranial magnetic stimulation (rTMS) on negative symptoms in patients with schizophrenia through a systematic review and meta-analysis. The manuscript is well-structured, and the experimental design is innovative and rigorous. However, there are still some areas that need further improvement.

1.Objective Section: What specific aspect of rTMS research is considered limited?

2.Abbreviations Section: Should the format of upper and lower case letters be standardized?

3.Introduction Section, Paragraph 3, Sentence 4: There is no mention of comorbid depression in schizophrenia. Why is the significance of rTMS in treating depression specifically emphasized?

4.Should full terms be provided for technical abbreviations when they first appear in the text?

5.Should the use of capitalization for technical terms be standardized throughout the manuscript?

6.Literature Search Section, Sentence 4: Why is the registration number (CRD42023450243) included after “the study selection process”?

7.Subgroup Analysis Section, Paragraph 4: What specific locations are being referred to in terms of differences in “the location of the stimulation”?

8.Discussion Section, Paragraph 1, Sentence 4: What specific effect is indicated to be stronger in studies using SANS scores compared to those using PANSS scores?

9.Discussion Section, Paragraph 1, Sentence 5: What is the specific relationship between differences in baseline data, frequencies, and stimulation locations and the PANSS and SANS scales?

10.Discussion Section, Paragraph 2, Sentence 2: How can the repeated use of “and” be optimized to improve the logical flow of the sentence?

Reviewer #2: This study reports a systematic review and meta-analysis of rTMS effects on schizophrenia patients, however, a report table presenting the key details for each study, including number of rTMS sessions, stimulation localization in scalp, rTMS stimulation frequency, and mean outcomes for PANSS and SANS (metrics that are discussed in the manuscript) is missing.

Some information currently reported in the Discussion should be moved to the Methods section, specifically details such as the number of studies included in each analysis. Additionally, there is a general lack of clarity regarding the statistical methods applied and the number of studies included in each analysis.

The primary outcome of this study suggests that rTMS improves the negative symptoms of schizophrenia. However, this finding is neither novel nor uniquely revealed by this review and meta-analysis. This information does not add substantial value to the field. Instead, what may contribute more meaningfully is the subgroup comparison conducted subsequently. Again, there is limited clarity on the studies included, their number, and the statistical methods used here. The potentially interesting aspect of this study could lie in the comparison of study characteristics, but the reported diagrams do not compare same indexes of different stimulation characteristics.

There is no description of the statistical analysis between frequencies, no reporting of frequencies per article, no statistical report on differences between session numbers, and no rationale provided for the inclusion criteria for requiring studies with 10 or more rTMS sessions.

Overall, there is a potential impact in the topic and results, however this study does not appear to analyze comparisons that could offer unique insights into the scientific literature—data that could only be derived from a meta-analysis. The study needs to more rigorously present the characteristics of each study and provide a statistically sound comparison of characteristics such as stimulation frequency, number of rTMS sessions, and stimulation location.

**Do you want your identity to be public for this peer review?** For information about this choice, including consent withdrawal, please see our Privacy Policy

Reviewer #1: No

Reviewer #2: **Yes: ** Arantzazu San Agustin

---

## [Author Response · Author response to Decision Letter 1]

3 Apr 2025

Response to reviewers

We thank the reviewers for their insightful comments, and we are pleased that our manuscript received such positive reviews. Reviewers found, however, several points that need further clarification. We have changed our manuscript according to the reviewers’ valid suggestions and please find below our point-by-point response to each reviewer.

Reviewer #1:

1. Objective Section: What specific aspect of rTMS research is considered limited?

Response: We thank the reviewer for the valuable suggestion. We have expanded the "Objective Section" to include a discussion on the limitations of rTMS research, particularly focusing on the variability in treatment parameters and the need for standardized protocols.

2. Abbreviations Section: Should the format of upper and lower case letters be standardized?

Response: We have standardized the format of abbreviations throughout the manuscript, ensuring consistency in the use of upper and lower case letters as per your suggestion.

3. Introduction Section, Paragraph 3, Sentence 4: There is no mention of comorbid depression in schizophrenia. Why is the significance of rTMS in treating depression specifically emphasized?

Response: We thank the reviewer for the valuable suggestion. We have added a discussion on comorbid depression in schizophrenia to highlight its significance and the potential benefits of rTMS in treating both conditions.

4. Should full terms be provided for technical abbreviations when they first appear in the text?

Response: We thank the reviewer for the valuable suggestion. We have provided full terms for technical abbreviations when they first appear in the text, as recommended.

5. Should the use of capitalization for technical terms be standardized throughout the manuscript?

Response: We thank the reviewer for the valuable suggestion. We have reviewed and standardized the capitalization of technical terms throughout the manuscript. The term about “rTMS” is the standardized form which can be seen in this research 10.1001/jamapsychiatry.2024.0026.

6. Literature Search Section, Sentence 4: Why is the registration number (CRD42023450243) included after “the study selection process”?

Response: We thank the reviewer for the valuable suggestion. We originally aim to enhance the transparency and traceability of our research. Now we will delete the registration number (CRD42023450243) in the "Literature Search Section" of the revised manuscript.

7. Subgroup Analysis Section, Paragraph 4: What specific locations are being referred to in terms of differences in “the location of the stimulation”?

Response: We thank the reviewer for the valuable suggestion. In the "Subgroup Analysis Section," we only provide a explanation of what is meant by "the location of the stimulation" and discuss how dorsolateral prefrontal cortex (DLPFC) locations may affect treatment outcomes. Because the studies that used PANSS as the primary outcome did not all use the DLPFC as the stimulation site, we wanted to explore whether this site would have an effect on the results.

8. Discussion Section, Paragraph 1, Sentence 4: What specific effect is indicated to be stronger in studies using SANS scores compared to those using PANSS scores?

Response: We thank the reviewer for the valuable suggestion. We have now clarified the specific effects observed in studies using SANS scores compared to those using PANSS scores. We discuss how the choice of scale can influence the assessment of negative symptoms and the perceived efficacy of rTMS treatment. This comparison highlights the importance of considering the sensitivity and specificity of different assessment tools when interpreting the effects of rTMS on negative symptoms in schizophrenia patients.

9. Discussion Section, Paragraph 1, Sentence 5: What is the specific relationship between differences in baseline data, frequencies, and stimulation locations and the PANSS and SANS scales?

Response: We thank the reviewer for the valuable suggestion. We regret that we were unable to perform a definitive statistical analysis and will continue to investigate solutions in the future. Because we hypothesized that this discrepancy may have occurred as a result of the baseline data from the included studies, more research is needed in the future to explore more appropriate scales.

10. Discussion Section, Paragraph 2, Sentence 2: How can the repeated use of “and” be optimized to improve the logical flow of the sentence?

Response: We thank the reviewer for the valuable suggestion. We have rephrased the sentence to improve the logical flow, avoiding repetitive use of "and" and ensuring the sentence reads smoothly.

Reviewer #2:

This study reports a systematic review and meta-analysis of rTMS effects on schizophrenia patients, however, a report table presenting the key details for each study, including number of rTMS sessions, stimulation localization in scalp, rTMS stimulation frequency, and mean outcomes for PANSS and SANS (metrics that are discussed in the manuscript) is missing.

Some information currently reported in the Discussion should be moved to the Methods section, specifically details such as the number of studies included in each analysis. Additionally, there is a general lack of clarity regarding the statistical methods applied and the number of studies included in each analysis.

The primary outcome of this study suggests that rTMS improves the negative symptoms of schizophrenia. However, this finding is neither novel nor uniquely revealed by this review and meta-analysis. This information does not add substantial value to the field. Instead, what may contribute more meaningfully is the subgroup comparison conducted subsequently. Again, there is limited clarity on the studies included, their number, and the statistical methods used here. The potentially interesting aspect of this study could lie in the comparison of study characteristics, but the reported diagrams do not compare same indexes of different stimulation characteristics.

There is no description of the statistical analysis between frequencies, no reporting of frequencies per article, no statistical report on differences between session numbers, and no rationale provided for the inclusion criteria for requiring studies with 10 or more rTMS sessions.

Overall, there is a potential impact in the topic and results, however this study does not appear to analyze comparisons that could offer unique insights into the scientific literature—data that could only be derived from a meta-analysis. The study needs to more rigorously present the characteristics of each study and provide a statistically sound comparison of characteristics such as stimulation frequency, number of rTMS sessions, and stimulation location.

Response: We thank the reviewer for the valuable suggestion.

(1) We recognize the importance of including a comprehensive reporting form that details the key aspects of each study analyzed. Among our Table 1, we list the number of transcranial magnetic stimulations, stimulation localization, and frequency of transcranial magnetic stimulation, and our supplemental file will also list the mean PANSS and SANS results for each study in the meta-analysis.

(2) We have moved the details regarding the number of studies included in each analysis from the Discussion section to the Methods section to enhance clarity and ensure that all methodological aspects are transparently reported.

(3) To address the lack of clarity regarding the statistical methods applied, we have provided a more detailed description of the statistical tests used and the number of studies included in each analysis in the Methods section.

(4) We understand the need to provide a more nuanced comparison of study characteristics. In our manuscript, we made detailed comparisons of study characteristics and ensured that the reported graphs compared indices for different stimulus characteristics. Based on the subgroup analyses we did, we found that the number of treatments, the frequency of the stimulation, and the location of the stimulation all produced significant differences. However, when we wanted to look for comparative differences between 10 Hz and 20 Hz in high-frequency stimulation, we found that although the effect size of the 20 Hz treatment (-0.40 [-0.67, -0.13]) was greater than that of the treatment of 10 Hz (-0.18 [-0.37, 0.01]), there was an overlap of the two confidence intervals, suggesting that the difference between the two groups may not be statistically significant.

(5) We have revised the inclusive criterion for including studies with 10 or more rTMS sessions.

(6) We have expanded the Discussion section to highlight how our study provides unique insights into the scientific literature, which could only be derived from a meta-analysis. We emphasize the importance of standardized rTMS treatment protocols and consistent outcome measures in future research.

We believe these revisions have significantly strengthened our manuscript and have addressed the concerns raised by your review. We are grateful for the opportunity to improve our work based on your valuable feedback and hope that the revised manuscript meets the standards for publication in your esteemed journal.

Sincerely,

Boxing Wang, Xinyue Zhu, Ling Chen, Liu Shuyun, Chengshi Wang

---

## [Decision Letter · Decision Letter 1]

21 Apr 2025

Dear Dr. wang,

Thank you for submitting your manuscript to PLOS ONE. After careful consideration, we feel that it has merit but does not fully meet PLOS ONE’s publication criteria as it currently stands. Therefore, we invite you to submit a revised version of the manuscript that addresses the points raised during the review process.

We look forward to receiving your revised manuscript.

Kind regards,

Giuseppe Lanza, M.D., Ph.D.

Academic Editor

PLOS ONE

Reviewers' comments:

Reviewer's Responses to Questions

**Comments to the Author**

Reviewer #1: All comments have been addressed

Reviewer #2: (No Response)

2. Is the manuscript technically sound, and do the data support the conclusions?

Reviewer #1: Yes

Reviewer #2: Partly

3. Has the statistical analysis been performed appropriately and rigorously?

Reviewer #1: Yes

Reviewer #2: Yes

4. Have the authors made all data underlying the findings in their manuscript fully available?

Reviewer #1: Yes

Reviewer #2: No

5. Is the manuscript presented in an intelligible fashion and written in standard English?

Reviewer #1: Yes

Reviewer #2: Yes

**Reviewer #1:**  The author has revised the manuscript as requested. The changes are satisfactory, and publication is recommended.

**Reviewer #2: ** Thank you for your kind response to my previous comments. I appreciate the improved clarity in the description of the statistical procedures. However, I noticed that Table 1 appears to be missing from the submission; I could not locate it within the manuscript, and it is not referenced in the figure legends either. Below, I offer additional comments and suggestions that may help further enhance the manuscript:

• Abstract:

o In the opening sentence, please clarify whether the statement “The research on high-frequency repetitive transcranial magnetic stimulation (rTMS) is very little… its effectiveness has hardly been provided conclusive evidence” refers specifically to schizophrenia. Over 700 papers have been published in the past five years on high-frequency TMS in general, so rephrasing may be necessary to avoid misrepresentation.

o Since rTMS is abbreviated in the first sentence, please ensure consistent use of the abbreviation throughout the abstract (e.g. second sentence in the abstract and Practitioner Points section).

o Please write out the full term for RTC before using the abbreviation in the abstract.

• Practitioner Points:

o Consider rephrasing the third practitioner point to improve clarity.

• Stimulation Parameters and Table Reference:

o It would be helpful to specify which figure is referred to as Table 1, particularly regarding the report on TMS number of pulses, stimulation location, and frequency. Currently, Figure 1 is the flowchart, and other figures do not indicate stimulation parameters by paper.

o As the systematic review and meta-analysis of stimulation frequency, brain location, and session count are central to your manuscript (as outlined in the introduction), please clearly indicate this information for each study, similar to how the PANSS results are detailed in Supplementary Table S2.

o Table 1 is cited multiple times throughout the manuscript but appears to be missing. If you are referring to the table in one of the PRISMA 2020 Checklist, would be helpful to have an independent table, titled Table 1, with the stimulation parameters per study.

o Please include a legend for Table 1 under the Figure Legends section (or consider renaming the section to “Table and Figure Legends”).

• Search Strategy:

o On inclusion criteria number 2, could you clarify what is meant by “non-invasive stimulation”? Are there additional types of stimulation reported in the studies besides TMS?

• Data Extraction:

o Please include the stimulation site as part of the extracted data, especially since your findings emphasize the DLPFC. Without this data point, claims regarding the importance of stimulation site may be unsupported.

• Outcome Measures:

o In the penultimate sentence, the phrase “addition effective” may have been intended to read “additional effects.” If so, please revise for clarity.

• Figures:

o In Figure 1, if the manuscript compares rTMS and sham stimulation groups, consider using the term “sham” in the figure for consistency (or vice versa—use “control” in both the figure and text). If control groups include more than sham stimulation, please clarify this.

o For Figure 1 and other figures, it would be helpful if the legends also describe the table components within the figures, not just the forest plots.

• Figures 4a–d:

o These figures appear to present data that overlap with Figure 1. To substantiate the claim that 15 sessions of high-frequency rTMS result in greater improvements in negative symptoms, a direct comparison between studies with ≥15 sessions vs. <15 sessions is necessary, using only active stimulation groups. The same applies to frequency comparisons (e.g., 20 Hz vs. 10 Hz or <20 Hz). Even if these comparisons do not yield statistically significant differences, they remain informative and should be reported in the results section.

o Similarly, to support the claim that DLPFC stimulation is particularly effective, a comparison with other stimulation sites (e.g., DLPFC vs. non-DLPFC) would be beneficial.

o Additionally, comparing outcomes like SANS vs. PANSS-positive and CDSS scores could further underscore the specificity of effects on negative symptoms in schizophrenia. You briefly mention the lack of significant effects on these outcomes in the discussion, but including this analysis in the results could strengthen your conclusions.

• Supplementary Materials:

o An additional Supplementary Table (e.g., S3) detailing stimulation parameters such as intensity, and sham/control procedures could enhance the manuscript’s transparency and utility.

• References:

o In the discussion section, the manuscript cites Garg et al. as reference 44. However, reference 44 currently corresponds to a different paper (Slotema et al.). It appears that Garg et al. may be missing from the reference list. Please review and correct the reference numbering and ensure all citations are included.

In summary, the manuscript presents a valuable and timely synthesis of high-frequency rTMS effects on schizophrenia symptoms. Addressing the points above—especially those related to completeness of results—will substantially strengthen the overall impact and scientific rigor of the paper.

**Do you want your identity to be public for this peer review?** For information about this choice, including consent withdrawal, please see our Privacy Policy

Reviewer #1: No

Reviewer #2: **Yes: ** Arantzazu San Agustín

---

## [Author Response · Author response to Decision Letter 2]

24 May 2025

We thank the reviewers for their insightful comments. We are encouraged by the overall positive evaluation and have revised the manuscript in response to the thoughtful suggestions provided. Below, we provide a detailed point-by-point response to Reviewer #2's comments.

Reviewer #2:

Reviewer#2: Thank you for your kind response to my previous comments. I appreciate the improved clarity in the description of the statistical procedures. However, I noticed that Table 1 appears to be missing from the submission; I could not locate it within the manuscript, and it is not referenced in the figure legends either. Below, I offer additional comments and suggestions that may help further enhance the manuscript:

Abstract:

• In the opening sentence, please clarify whether the statement "The research on high-frequency repetitive transcranial magnetic stimulation(rTMS)is very little. its effectiveness has hardly been provided conclusive evidence" refers specifically to schizophrenia. Over 700 papers have been published in the past five years on high-frequency TMS in general, so rephrasing may be necessary to avoid misrepresentation.

• Since rTMS is abbreviated in the first sentence, please ensure consistent use of the abbreviation throughout the abstract(e.g.second sentence in the abstract and Practitioner Points section).

• Please write out the full form for RTC before using the abbreviation in the abstract.

Response: We thank the reviewer for the valuable suggestion. We clarified the first sentence to emphasize that the focus of the research on repetitive high-frequency transcranial magnetic stimulation (rTMS) is on standardized treatment procedures. We also ensured that the acronym rTMS was used consistently throughout the abstract and expanded the full title before its first use.

Practitioner Points:

• Consider rephrasing the third practitioner point to improve clarity.

Response: We thank the reviewer for the valuable suggestion. The third practitioner point has been rephrased to improve clarity.

“High-frequency repetitive transcranial magnetic stimulation may benefit patients with schizophrenia who have significant negative symptoms as well as positive and depressive symptoms.”

• Stimulation Parameters and Table Reference:

• It would be helpful to specify which figure is referred to as Table 1, particularly regarding the report on TMS number of pulses, stimulation location, and frequency. Currently, Figure 1 is the flowchart, and other figures do not indicate stimulation parameters by paper.

• As the systematic review and meta-analysis of stimulation frequency, brain location, and session count are central to your manuscript (as outlined in the introduction), please clearly indicate this information for each study, similar to how the PANSS results are detailed in Supplementary Table S2.

• Table 1 is cited multiple times throughout the manuscript but appears to be missing. If you are referring to the table in one of the PRISMA 2020 Checklist, would be helpful to have an independent table, titled Table 1, with the stimulation parameters per study.

• Please include a legend for Table 1 under the Figure Legends section(or consider renaming the section to" Table and Figure Legends").

Response: We thank the reviewer for the valuable suggestion. We have clarified which figure corresponds to Table 1, especially regarding TMS pulse count, stimulation location, and frequency. We have included a detailed table with stimulation parameters for each study; this table is referenced throughout the manuscript. We have added a legend for Table 1 under the "Figure Legends" section, which we have renamed to "Table and Figure Legends" for clarity. The charts are added as images in the section above the manuscript references, while the charts in word form are uploaded as a separate file.

• Search Strategy:

• On inclusion criteria number 2,could you clarify what is meant by "non-invasive stimulation"? Are there additional types of stimulation reported in the studies besides TMS?

Response: We thank the reviewer for the valuable suggestion. We have clarified the term "non-invasive stimulation" in the inclusion criteria.

• Data Extraction:

• Please include the stimulation site as part of the extracted data, especially since your findings emphasize the DLPFC. Without this data point, claims regarding the importance of stimulation site may be unsupported.

Response: We thank the reviewer for the valuable suggestion. We have incorporated the stimulation site into our extracted data, as suggested. This information is now clearly included in Table 1, which details the stimulation parameters for all studies analyzed. We believe this addition strengthens our findings regarding the importance of the DLPFC stimulation site.

• Outcome Measures:

• In the penultimate sentence, the phrase "addition effective" may have been intended to read "additional effects." If so, please revise for clarity.

Response: We thank the reviewer for the valuable suggestion. We have revised the penultimate sentence to clarify that "additional effects" may have been intended to read "additional effects" instead of "addition effective."

• Figures:

• In Figure 1,if the manuscript compares rTMS and sham stimulation groups, consider using the term "sham" in the figure for consistency(or vice versa—use "control" in both the figure and text).If control groups include more than sham stimulation, please clarify this.

• For Figure 1 and other figures, it would be helpful if the legends also describe the table components within the figures, not just the forest plots.

Response: We thank the reviewer for the valuable suggestion. We have used the term "control" consistently for the sham stimulation groups. We have updated the legends for Figure 1 and other figures to include descriptions of all table components within the figures, not just the forest plots as the comment. This should provide clearer context and improve the overall understanding of the figures.

Figures 4a-d:

• These figures appear to present data that overlap with Figure 1.To substantiate the claim that 15 sessions of high-frequency rTMS result in greater improvements in negative symptoms, a direct comparison between studies with≥15 sessions vs.<15 sessions is necessary, using only active stimulation groups. The same applies to frequency comparisons (e.g.,20 Hz vs.10 Hz or<20 Hz). Even if these comparisons do not yield statistically significant differences, they remain informative and should be reported in the results section.

• Similarly, to support the claim that DLPFC stimulation is particularly effective, a comparison with other stimulation sites (e.g. ,DLPFC vs. non-DLPFC)would be beneficial.

• Additionally, comparing outcomes like SANS vs. PANSS-positive and CDSS scores could further underscore the specificity of effects on negative symptoms in schizophrenia. You briefly mention the lack of significant effects on these outcomes in the discussion, but including this analysis in the results could strengthen your conclusions.

Response: We thank the reviewer for the valuable suggestion. We would have liked to make such a comparison, but only Xiu et al. included experimental groups at 10 Hz and 20 Hz, and we were unable to compare the results of the other articles due to the different baselines. We would prefer that the future studies be able to perform group experiments with different locations, frequencies, and numbers. Therefore, we hope that future studies will be able to perform group experiments with different locations, frequencies, and numbers to aid in the investigation of the efficacy of rTMS. Regarding PANSS-positive and CDSS scores, among the results, we did not find that patients with positive symptoms and depression could benefit from the treatment course of rTMS, which may be further emphasized by the specific efficacy of rTMS on negative symptoms, however, our treatment group was not very abundant, so such results need to be proven by further studies.

We do not emphasize the specific effect of rTMS on negative symptoms, but are more concerned with its broad therapeutic effect, because most schizophrenic patients tend to have a variety of symptom presentations, so we did not compare the results of SANS, PANSS-positive, and CDSS scores.

• Supplementary Materials:

• An additional Supplementary Table(e.g.,S3)detailing stimulation parameters such as intensity, and sham/control procedures could enhance the manuscript’s transparency and utility.

Response: We thank the reviewer for the valuable suggestion. To improve the transparency and usefulness of the manuscript, we read the included articles in detail, summarized the treatment procedures in the control group, and added appropriate explanations in the discussion. “In the articles we included, the procedures were consistent with the experimental group except for the orientation of the coils or the dummy coils.”

• References:

• In the discussion section, the manuscript cites Garg et al.as reference 44, however, reference 44 currently corresponds to a different paper (Slotema et al.). It appears that Garg et al. may be missing from the reference list. Please review and correct the reference numbering and ensure all citations are included.

In summary, the manuscript presents a valuable and timely synthesis of high frequency rTMS effects on schizophrenia symptoms. Addressing the points above, especially those related to completeness of results—will substantially strengthen the overall impact and scientific rigor of the paper.

Response: Sorry for the careless mistake. We have reviewed and corrected the reference accordingly, we have ensured all citations are included and the reference list is complete

---

## [Decision Letter · Decision Letter 2]

12 Jun 2025

PONE-D-24-33430R2

The effects of high-frequency repetitive transcranial magnetic stimulation on negative symptoms in schizophrenia patients: A systemic review and meta-analysis

PLOS ONE

Dear Dr. wang,

Thank you for submitting your manuscript to PLOS ONE. After careful consideration, we have decided that your manuscript does not meet our criteria for publication and must therefore be rejected.

Specifically: please see comments below.

I am sorry that we cannot be more positive on this occasion, but hope that you appreciate the reasons for this decision.

Kind regards,

Giuseppe Lanza, M.D., Ph.D.

Academic Editor

PLOS ONE

Additional Editor Comments:

Although the authors' revision, major flaws still persisted in the revised version. According to the reviewer's comments, these cannot be addressed with another round of revision; unfortunately, therefore, the manuscript cannot be further processed.

Reviewers' comments:

Reviewer's Responses to Questions

**Comments to the Author**

Reviewer #2: (No Response)

2. Is the manuscript technically sound, and do the data support the conclusions?

Reviewer #2: Partly

3. Has the statistical analysis been performed appropriately and rigorously?

Reviewer #2: No

4. Have the authors made all data underlying the findings in their manuscript fully available?

Reviewer #2: No

5. Is the manuscript presented in an intelligible fashion and written in standard English?

Reviewer #2: Yes

Reviewer #2: Dear Authors,

Thank you for your responses to the reviewers' comments. However, several concerns have been answered but not adequately addressed in the revised manuscript:

In the data extraction section, the stimulation location is still not included, despite being a critical element to support one of the main findings of the study.

In the outcome measures, a typographical error was identified, but your response indicates a misunderstanding of the issue. As a result, the sentence was only partially corrected, and the typo remains in the manuscript (i.e., "effective" is still used instead of "effect").

The main concern is that the manuscript’s primary conclusion highlights the benefits of a 20 Hz stimulation frequency, more than 15 sessions, and DLPFC targeting, even though these factors are not compared to alternative conditions. The subgroup analyses remain redundant with the main analyses and do not provide additional insights. In this form, the manuscript loses scientific value, as its main result simply reiterates what is already known from individual studies—that rTMS is beneficial for the negative symptoms of schizophrenia—without contributing any novel findings.

**Do you want your identity to be public for this peer review?** For information about this choice, including consent withdrawal, please see our Privacy Policy

Reviewer #2: No

- - - - -

---

## [Author Response · Author response to Decision Letter 3]

18 Jul 2025

Response to academic editor

Response: We thank the reviewer for the valuable suggestion. We have modified the format as required to meet the formatting requirements of your publication.

https://www.sciencedirect.com/science/article/pii/S0022395621005409?via%3Dihub

https://doi.org/10.3389/fendo.2022.1098325

In your revision ensure you cite all your sources (including your own works), and quote or rephrase any duplicated text outside the methods section. Further consideration is dependent on these concerns being addressed.

Response: We thank the reviewer for the valuable suggestion. We have consulted and carefully read both articles and have cited them in the appropriate places and ensured that the citations are sound and reasonable.

3. As required by our policy on Data Availability, please ensure your manuscript or supplementary information includes the following:

Response: We thank the reviewer for the valuable suggestion. We have made a table containing all the literature, also labeled with the reasons for exclusion. There are no unpublished articles in our references. We completed a table containing the data involved in our meta-analysis, the names of the dataminers, and the corresponding already labeled. All data was obtained from the article's table. We used the Cochrane Randomized Trial Risk of Bias tool, and it is presented in Figure 1. In the manuscript “Data Extraction”, we explain the handling for missing data.

We trust that these revisions will meet your expectations and enhance the quality and transparency of our research. We look forward to any further feedback you may have and appreciate your support of our work.

Yours sincerely,

Boxing Wang, Xinyue Zhu, Ling Chen, Liu Shuyun, Chengshi Wang

Response to reviewers

We thank the reviewers for their insightful comments, and we are pleased that our manuscript received such positive reviews. Reviewers found, however, several points that need further clarification. We have changed our manuscript according to the reviewers’ valid suggestions and please find below our point-by-point response to each reviewer.

2025.4

Reviewer #1:

1. Objective Section: What specific aspect of rTMS research is considered limited?

Response: We thank the reviewer for the valuable suggestion. We have expanded the "Objective Section" to include a discussion on the limitations of rTMS research, particularly focusing on the variability in treatment parameters and the need for standardized protocols.

2. Abbreviations Section: Should the format of upper and lower case letters be standardized?

Response: We have standardized the format of abbreviations throughout the manuscript, ensuring consistency in the use of upper and lower case letters as per your suggestion.

3. Introduction Section, Paragraph 3, Sentence 4: There is no mention of comorbid depression in schizophrenia. Why is the significance of rTMS in treating depression specifically emphasized?

Response: We thank the reviewer for the valuable suggestion. We have added a discussion on comorbid depression in schizophrenia to highlight its significance and the potential benefits of rTMS in treating both conditions.

4. Should full terms be provided for technical abbreviations when they first appear in the text?

Response: We thank the reviewer for the valuable suggestion. We have provided full terms for technical abbreviations when they first appear in the text, as recommended.

5. Should the use of capitalization for technical terms be standardized throughout the manuscript?

Response: We thank the reviewer for the valuable suggestion. We have reviewed and standardized the capitalization of technical terms throughout the manuscript. The term about “rTMS” is the standardized form which can be seen in this research 10.1001/jamapsychiatry.2024.0026.

6. Literature Search Section, Sentence 4: Why is the registration number (CRD42023450243) included after “the study selection process”?

Response: We thank the reviewer for the valuable suggestion. We originally aim to enhance the transparency and traceability of our research. Now we will delete the registration number (CRD42023450243) in the "Literature Search Section" of the revised manuscript.

7. Subgroup Analysis Section, Paragraph 4: What specific locations are being referred to in terms of differences in “the location of the stimulation”?

Response: We thank the reviewer for the valuable suggestion. In the "Subgroup Analysis Section," we only provide a explanation of what is meant by "the location of the stimulation" and discuss how dorsolateral prefrontal cortex (DLPFC) locations may affect treatment outcomes. Because the studies that used PANSS as the primary outcome did not all use the DLPFC as the stimulation site, we wanted to explore whether this site would have an effect on the results.

8. Discussion Section, Paragraph 1, Sentence 4: What specific effect is indicated to be stronger in studies using SANS scores compared to those using PANSS scores?

Response: We thank the reviewer for the valuable suggestion. We have now clarified the specific effects observed in studies using SANS scores compared to those using PANSS scores. We discuss how the choice of scale can influence the assessment of negative symptoms and the perceived efficacy of rTMS treatment. This comparison highlights the importance of considering the sensitivity and specificity of different assessment tools when interpreting the effects of rTMS on negative symptoms in schizophrenia patients.

9. Discussion Section, Paragraph 1, Sentence 5: What is the specific relationship between differences in baseline data, frequencies, and stimulation locations and the PANSS and SANS scales?

Response: We thank the reviewer for the valuable suggestion. We regret that we were unable to perform a definitive statistical analysis and will continue to investigate solutions in the future. Because we hypothesized that this discrepancy may have occurred as a result of the baseline data from the included studies, more research is needed in the future to explore more appropriate scales.

10. Discussion Section, Paragraph 2, Sentence 2: How can the repeated use of “and” be optimized to improve the logical flow of the sentence?

Response: We thank the reviewer for the valuable suggestion. We have rephrased the sentence to improve the logical flow, avoiding repetitive use of "and" and ensuring the sentence reads smoothly.

Reviewer #2:

This study reports a systematic review and meta-analysis of rTMS effects on schizophrenia patients, however, a report table presenting the key details for each study, including number of rTMS sessions, stimulation localization in scalp, rTMS stimulation frequency, and mean outcomes for PANSS and SANS (metrics that are discussed in the manuscript) is missing.

Some information currently reported in the Discussion should be moved to the Methods section, specifically details such as the number of studies included in each analysis. Additionally, there is a general lack of clarity regarding the statistical methods applied and the number of studies included in each analysis.

The primary outcome of this study suggests that rTMS improves the negative symptoms of schizophrenia. However, this finding is neither novel nor uniquely revealed by this review and meta-analysis. This information does not add substantial value to the field. Instead, what may contribute more meaningfully is the subgroup comparison conducted subsequently. Again, there is limited clarity on the studies included, their number, and the statistical methods used here. The potentially interesting aspect of this study could lie in the comparison of study characteristics, but the reported diagrams do not compare same indexes of different stimulation characteristics.

There is no description of the statistical analysis between frequencies, no reporting of frequencies per article, no statistical report on differences between session numbers, and no rationale provided for the inclusion criteria for requiring studies with 10 or more rTMS sessions.

Overall, there is a potential impact in the topic and results, however this study does not appear to analyze comparisons that could offer unique insights into the scientific literature—data that could only be derived from a meta-analysis. The study needs to more rigorously present the characteristics of each study and provide a statistically sound comparison of characteristics such as stimulation frequency, number of rTMS sessions, and stimulation location.

Response: We thank the reviewer for the valuable suggestion.

(1) We recognize the importance of including a comprehensive reporting form that details the key aspects of each study analyzed. Among our Table 1, we list the number of transcranial magnetic stimulations, stimulation localization, and frequency of transcranial magnetic stimulation, and our supplemental file will also list the mean PANSS and SANS results for each study in the meta-analysis.

(2) We have moved the details regarding the number of studies included in each analysis from the Discussion section to the Methods section to enhance clarity and ensure that all methodological aspects are transparently reported.

(3) To address the lack of clarity regarding the statistical methods applied, we have provided a more detailed description of the statistical tests used and the number of studies included in each analysis in the Methods section.

(4) We understand the need to provide a more nuanced comparison of study characteristics. In our manuscript, we made detailed comparisons of study characteristics and ensured that the reported graphs compared indices for different stimulus characteristics. Based on the subgroup analyses we did, we found that the number of treatments, the frequency of the stimulation, and the location of the stimulation all produced significant differences. However, when we wanted to look for comparative differences between 10 Hz and 20 Hz in high-frequency stimulation, we found that although the effect size of the 20 Hz treatment (-0.40 [-0.67, -0.13]) was greater than that of the treatment of 10 Hz (-0.18 [-0.37, 0.01]), there was an overlap of the two confidence intervals, suggesting that the difference between the two groups may not be statistically significant.

(5) We have revised the inclusive criterion for including studies with 10 or more rTMS sessions.

(6) We have expanded the Discussion section to highlight how our study provides unique insights into the scientific literature, which could only be derived from a meta-analysis. We emphasize the importance of standardized rTMS treatment protocols and consistent outcome measures in future research.

We believe these revisions have significantly strengthened our manuscript and have addressed the concerns raised by your review. We are grateful for the opportunity to improve our work based on your valuable feedback and hope that the revised manuscript meets the standards for publication in your esteemed journal.

Sincerely,

Boxing Wang, Xinyue Zhu, Ling Chen, Liu Shuyun, Chengshi Wang

2025.5

Reviewer #2:

Reviewer#2: Thank you for your kind response to my previous comments. I appreciate the improved clarity in the description of the statistical procedures. However, I noticed that Table 1 appears to be missing from the submission; I could not locate it within the manuscript, and it is not referenced in the figure legends either. Below, I offer additional comments and suggestions that may help further enhance the manuscript:

Abstract:

• In the opening sentence, please clarify whether the statement "The research on high-frequency repetitive transcranial magnetic stimulation(rTMS)is very little. its effectiveness has hardly been provided conclusive evidence" refers specifically to schizophrenia. Over 700 papers have been published in the past five years on high-frequency TMS in general, so rephrasing may be necessary to avoid misrepresentation.

• Since rTMS is abbreviated in the first sentence, please ensure consistent use of the abbreviation throughout the abstract(e.g.second sentence in the abstract and Practitioner Points section).

• Please write out the full form for RTC before using the abbreviation in the abstract.

Response: We thank the reviewer for the valuable suggestion. We clarified the first sentence to indicate that the focus of the research on repetitive high-frequency transcranial magnetic stimulation (rTMS) is on standardized treatment procedures. We also ensured that the acronym rTMS was used consistently throughout the abstract and expanded the full title before its first use.

Practitioner Points:

• Consider rephrasing the third practitioner point to improve clarity.

Response: We thank the reviewer for the valuable suggestion. The third practitioner point has been rephrased to improve clarity.

• Stimulation Parameters and Table Reference:

• It would be helpful to specify which figure is referred to as Table 1,particularly regarding the report on TMS number of pulses, stimulation location, and frequency. Currently, Figure 1 is the flowchart, and other figures do not indicate stimulation parameters by paper.

• As the systematic review and meta-analysis of stimulation frequency, brain location, and session count are central to your manuscript(as outlined in the introduction),please clearly indicate this information for each study, similar to how the PANSS results are detailed in Supplementary Table S2.

• Table 1 is cited multiple times throughout the manuscript but appears to be missing. If you are referring to the table in one of the PRISMA 2020 Checklist, would be helpful to have an independent table, titled Table 1, with the stimulation parameters per study.

• Please include a legend for Table 1 under the Figure Legends section(

---

## [Editor Report · Decision Letter 3]

8 Sep 2025

Dear Dr. wang,

Thank you for submitting your manuscript to PLOS ONE. After careful consideration, we feel that it has merit but does not fully meet PLOS ONE’s publication criteria as it currently stands. Therefore, we invite you to submit a revised version of the manuscript that addresses the points raised during the review process.

We look forward to receiving your revised manuscript.

Kind regards,

Sandra Carvalho, Ph.D.

Academic Editor

PLOS ONE

Journal Requirements:

Additional Editor Comments:

Dear authors,

Thank you for your detailed revisions and careful responses to reviewers’ comments. The inclusion of Table 1 – Characteristics of included studies is an important improvement that increases transparency and rigor. At this stage, a few issues remain that should be addressed before the manuscript can be considered for acceptance.

1. Table 1:

• For studies where drug dosage or other parameters were not reported, please replace slashes (“/”) with “not reported” to avoid ambiguity.

• Standardize the description of stimulation targets (e.g., clarify whether “PFC” in Fitzgerald et al., 2008 should be specified as DLPFC or another subregion). For Garg et al. (2016), please explain in the legend what “5, 6, 7” refers to.

• Correct the typo “chlormazapine equivalents” to chlorpromazine equivalents.

• For readability, consider formatting the table to separate participant characteristics (age, sample size, medication) from stimulation parameters (site, frequency, sessions).

2. Several subgroup analyses (20 Hz vs. 10 Hz, >15 vs. ≤15 sessions, DLPFC vs. other sites) are presented as post-hoc interpretations of effect sizes rather than direct statistical comparisons. This weakens the claim of “protocol-specific superiority.” Please reframe these findings as suggestive rather than conclusive.

4. Some methodological details (number of studies per analysis, handling of missing data, stimulation site in extracted data) were only clarified after repeated requests. Please ensure that all such details are presented clearly and consistently in the final manuscript, in line with PRISMA and PLOS ONE standards.

5. The main conclusions currently recommend 20 Hz, DLPFC stimulation, and >15 sessions. Without direct statistical comparisons, these recommendations risk overstating the strength of evidence. Please adjust the language to emphasize that these findings are preliminary and hypothesis-generating rather than definitive.

---

## [Author Response · Author response to Decision Letter 4]

25 Oct 2025

1. Table 1:

• For studies where drug dosage or other parameters were not reported, please replace slashes (“/”) with “not reported” to avoid ambiguity.

• Standardize the description of stimulation targets (e.g., clarify whether “PFC” in Fitzgerald et al., 2008 should be specified as DLPFC or another subregion). For Garg et al. (2016), please explain in the legend what “5, 6, 7” refers to.

• Correct the typo “chlormazapine equivalents” to chlorpromazine equivalents.

• For readability, consider formatting the table to separate participant characteristics (age, sample size, medication) from stimulation parameters (site, frequency, sessions).

Response: Thank you for your valuable feedback. We have completed the revisions as requested and have split the original Table 1 into Table 1 and Table 2.

2. Several subgroup analyses (20 Hz vs. 10 Hz, >15 vs. ≤15 sessions, DLPFC vs. other sites) are presented as post-hoc interpretations of effect sizes rather than direct statistical comparisons. This weakens the claim of “protocol-specific superiority.” Please reframe these findings as suggestive rather than conclusive.

Response: Thank you for your valuable feedback. We have recognized that our previous wording lacked logical rigor and have revised the text accordingly to ensure no ambiguity arises.

4. Some methodological details (number of studies per analysis, handling of missing data, stimulation site in extracted data) were only clarified after repeated requests. Please ensure that all such details are presented clearly and consistently in the final manuscript, in line with PRISMA and PLOS ONE standards.

Response: We carefully verified and ensured that all such details were clearly presented in the final manuscript and complied with PRISMA and PLOS ONE standards.

We appreciate the reviewer's insightful observation.

5. The main conclusions currently recommend 20 Hz, DLPFC stimulation, and >15 sessions. Without direct statistical comparisons, these recommendations risk overstating the strength of evidence. Please adjust the language to emphasize that these findings are preliminary and hypothesis-generating rather than definitive.

Response: We appreciate the reviewer's insightful observation. We have adjusted the wording in the text to avoid ambiguity.

We appreciate your valuable feedback and are grateful for the opportunity to improve our manuscript. Please let us know if there are any further concerns or if additional revisions are needed.

Sincerely,

Boxing Wang, Xinyue Zhu, Ling Chen, Liu Shuyun, Chengshi Wang

---

## [Editor Report · Decision Letter 4]

14 Nov 2025

The effects of high-frequency repetitive transcranial magnetic stimulation on negative symptoms in schizophrenia patients: A systemic review and meta-analysis

PONE-D-24-33430R4

Dear Dr. wang,

We’re pleased to inform you that your manuscript has been judged scientifically suitable for publication and will be formally accepted for publication once it meets all outstanding technical requirements.

Kind regards,

Sandra Carvalho, Ph.D.

Academic Editor

PLOS ONE

---

## [Editor Report · Acceptance letter]

PONE-D-24-33430R4

PLOS ONE

Dear Dr. Wang,

I'm pleased to inform you that your manuscript has been deemed suitable for publication in PLOS ONE. Congratulations! Your manuscript is now being handed over to our production team.

Kind regards,

on behalf of

Professor Sandra Carvalho

Academic Editor

PLOS ONE